# Bayesian Robust Optimization for Imitation Learning

**Daniel S. Brown**\*
UC Berkeley
dsbrown@berkeley.edu

**Scott Niekum**
University of Texas at Austin
sniekum@cs.utexas.edu

**Marek Petrik**
University of New Hampshire
mpetrik@cs.unh.edu

## Abstract

One of the main challenges in imitation learning is determining what action an agent should take when outside the state distribution of the demonstrations. Inverse reinforcement learning (IRL) can enable generalization to new states by learning a parameterized reward function, but these approaches still face uncertainty over the true reward function and corresponding optimal policy. Existing safe imitation learning approaches based on IRL deal with this uncertainty using a maxmin framework that optimizes a policy under the assumption of an adversarial reward function, whereas risk-neutral IRL approaches either optimize a policy for the mean or MAP reward function. While completely ignoring risk can lead to overly aggressive and unsafe policies, optimizing in a fully adversarial sense is also problematic as it can lead to overly conservative policies that perform poorly in practice. To provide a bridge between these two extremes, we propose Bayesian Robust Optimization for Imitation Learning (BROIL). BROIL leverages Bayesian reward function inference and a user specific risk tolerance to efficiently optimize a robust policy that balances expected return and conditional value at risk. Our empirical results show that BROIL provides a natural way to interpolate between return-maximizing and risk-minimizing behaviors and outperforms existing risk-sensitive and risk-neutral inverse reinforcement learning algorithms.

## 1 Introduction

Imitation learning [40] aims to train an agent without hand-specifying a reward function by providing demonstrations. One of the main challenges in imitation learning is determining what action an agent should take when outside the states contained in the demonstrations. Inverse reinforcement learning (IRL) [38] is an approach to imitation learning in which the learning agent seeks to recover the reward function of the demonstrator. Learning a parameterized reward function provides a compact representation of the demonstrator's preferences and enables generalization to new states unseen in the demonstrations via policy optimization. However, IRL approaches still result in uncertainty over the true reward function and this uncertainty can have negative consequences if the learning agent infers a reward function that leads it to learn an incorrect policy. In this paper we propose that an imitation learning agent should learn a policy that is robust with respect to its uncertainty over the true objective of a task, but also be able to effectively trade-off epistemic risk with expected return.

For example, consider two scenarios: (1) an autonomous car detects a novel object lying in the road ahead of the car and (2) a domestic service robot tasked with vacuuming encounters a pattern on the floor it has never seen before. The first example concerns autonomous driving where the car's decisions have potentially catastrophic consequences. Thus, the car should treat the novel object as a hazard and either slow down or safely change lanes to avoid running into it. In the second example, vacuuming the floors of a house has certain risks, but the consequences of optimizing the wrong reward function are arguably much less significant. Thus, when the vacuuming robot encounters a novel floor pattern it does not need to worry as much about negative side-effects.

Risk-averse optimization, especially in financial domains, has a long history of seeking to address the trade-off between risk and return using measures of risk such as variance [37], value at risk [30] and conditional value at risk [48]. This work has been extended to risk-averse optimization in Markov decision processes [15, 43, 44] and in the context of reinforcement learning [23, 58, 59], where the transition dynamics and reward function are not known. However, there has only been limited work in applying techniques for trading off risk and return in the domain of imitation learning. Brown et al. [11] seek to bound the value at risk of a policy in the imitation learning setting; however, directly optimizing a policy for value at risk is NP-hard [16]. Lacotte et al. [33] and Majumadar et al. [35] assume that risk-sensitive trajectories are available from a safe demonstrator and seek to optimize a policy that matches the risk-profile of this expert. In contrast, our approach directly optimizes a policy that balances expected return and conditional value at risk [48] which can be done via convex optimization. Furthermore, we do not try to match the demonstrator's risk sensitivity, but instead find a robust policy with respect to uncertainty over the demonstrator's reward function, allowing us to optimize policies that are potentially safer than the demonstrations.

One of the concerns of imitation learning, and especially inverse reinforcement learning, is the possibility of learning an incorrect reward function that leads to negative side-effects, for example, a vacuuming robot that learns that it is good to vacuum up dirt, but then goes around making messes for itself to clean up [50]. To address negative side-effects, most prior work on safe inverse reinforcement learning takes a minmax approach and seeks to optimize a policy with respect to the worst-case reward function [25, 28, 57]; however, treating the world as if it is completely adversarial (e.g., completely avoiding a novel patch of red flooring because it could potentially be lava [25]) can lead to overly conservative behaviors. On the other hand, other work on inverse reinforcement learning and imitation learning takes a risk neutral approach and simply seeks to perform well in expectation with respect to uncertainty over the demonstrator's reward function [46, 65]. This can result in behaviors that are overly optimistic in the face of uncertainty and can lead to policies with high variance in performance which is undesirable in high-risk domains like medicine or autonomous driving. Instead of assuming either a purely adversarial environment or a risk-neutral one, we propose the first inverse reinforcement learning algorithm capable of appropriately balancing caution with expected performance in a way that reflects the risk-sensitivity of the particular application.

The main contributions of this work are: (1) We propose Bayesian Robust Optimization for Imitation Learning (BROIL), the first imitation learning framework to directly optimize a policy that balances the expected return and the conditional value at risk under an uncertain reward function; (2) We derive an efficient linear programming formulation to compute the BROIL optimal policy; (3) We propose and compare two instantiations of BROIL: optimizing a purely robust policy with respect to uncertainty and optimizing a policy that minimizes baseline regret with respect to expert demonstrations; and (4) We demonstrate that BROIL achieves better expected return and robustness than existing risk-sensitive and risk-neutral IRL algorithms, as well as providing a richer class of solutions that correctly balance performance and risk based on different levels of risk aversion.

## 2   Related Work

An important challenge in inverse reinforcement learning (IRL) is dealing with ambiguity over the reward function [38, 65], since there are usually an infinite number of reward functions that are consistent with a set of demonstrations [38]. Problems with such ambiguous parameters can be solved using *robust optimization* techniques, which compute the best policy for the worst rewards consistent with the demonstrations [6]. Indeed, many IRL methods optimize policies for the worst-case rewards [25, 27, 28, 57]. This optimization for the worst-case parameter values is well known to lead to overly conservative solutions across many domains [16, 29, 49]. Bayesian IRL infers a posterior distribution over which rewards are most likely, but often only optimize for the mean [46] or MAP [14] reward function. Instead of optimizing only for the mean or worst-case reward values, we optimize for the expected performance across uncertain rewards while ensuring acceptable performance with high confidence. We rely on coherent measures of risk to represent the trade-off between the average and worst-case performance [3, 21, 54]. Similar approaches to parameter uncertainty, also known as epistemic uncertainty, have been referred to as soft-robustness in earlier work [5, 18] but have not been studied in the context of IRL.

In Table 1, we summarize pertinent properties of IRL methods with a focus on robustness or risk-aversion. VaR-BIRL [10, 11], a closely-related method, uses VaR, a risk measure, to quantify

Table 1: Summary of differences between BROIL and related robust IRL algorithms.

| | **BROIL** (ours) | RS-GAIL [33] | VaR-BIRL [10, 11] | RBIRL [64] | FPL-IRL [28] | LPAL [57] | GAIL [27] |
|---|---|---|---|---|---|---|---|
| Bayesian robust criterion | ✓ | · | ✓ | · | · | · | · |
| Risk-averse expert | · | ✓ | · | · | · | · | · |
| Exploits Bayesian prior | ✓ | · | ✓ | ✓ | · | · | · |
| Robust to bad demos | · | · | · | ✓ | · | · | · |
| Baseline regret objective | ✓ | ✓ | ✓ | · | · | ✓ | ✓ |
| Optimizes policies | ✓ | ✓ | · | ✓ | ✓ | ✓ | ✓ |

the robustness of a given policy in Bayesian IRL. Unfortunately, extending VaR-BIRL to policy optimization is difficult since the problem of optimizing VaR in an MDP with uncertain rewards is NP-hard [16]. Additionally, VaR ignores both the tail-risk of the distribution, as well as its average value, which may be undesirable for highly risk-sensitive problems [48].

While some of the methods in Table 1 resemble our approach, they differ either in their focus or the approach. RS-GAIL and related algorithms [33, 35, 52] also mitigate risk in IRL but assume risk-averse experts and focus on optimizing policies that match the risk-aversion of the demonstrator. These methods focus on the uncertainty induced by transition probabilities, also known as aleatoric risk. The challenges in this area are very different and there is no obvious way to adapt risk-averse IRL to our Bayesian robust setting where we seek to be robust to epistemic risk rather than seeking to match the risk of the demonstrator. RBIRL [64] aims to infer a posterior distribution that is robust to small numbers of bad demonstrations, but does not address robust policy optimization with respect to ambiguity in the learned posterior. While not explicitly robust to bad demonstrations, our method makes use of any posterior distribution over reward functions and can easily be extended to use posteriors generated from methods like RBIRL [64]. Finally, FPL-IRL [28], LPAL [57], and GAIL [27] optimize the policy for a (regularized) worst-case realization of the rewards and do not attempt to balance it with the average performance.

Another important point of difference among robust IRL algorithms is the objectives they optimize for. For example, FPL-IRL [28] focuses on the absolute performance of a policy, while GAIL [22, 27] optimizes the regret (or loss) relative to the policy of the demonstrator. In reinforcement learning, it has been shown that optimizing the regret is more appropriate if a good baseline policy is available [32, 34, 43]. There are similar advantages to optimizing the regret for the optimal policy [1, 2, 47]. We would also like to emphasize that our setting is quite different from robust RL methods which focus on uncertain transition probabilities rather than rewards [20, 26, 49, 60, 62]. Unlike much robust RL work, the optimization problems we derive are tractable without requiring rectangularity assumptions [24, 36].

## 3 Preliminaries

Before describing our method in Section 4, we briefly introduce our notation and review some of the concepts necessary to understand our approach. We use uppercase boldface and lowercase boldface characters to denote matrices and vectors respectively.

### 3.1 Markov Decision Processes

We model the environment as a Markov Decision Process (MDP) [45]. An MDP is a tuple $(\mathcal{S}, \mathcal{A}, r, P, \gamma, p_0)$, where $\mathcal{S} = \{s_1, \ldots, s_S\}$ are the states, $\mathcal{A} = \{a_1, \ldots, a_A\}$ are the actions, $r : \mathcal{S} \times \mathcal{A} \to \mathbb{R}$ is the reward function, $P : \mathcal{S} \times \mathcal{A} \times \mathcal{S} \to \mathbb{R}$ is the transition function, $\gamma \in [0, 1)$ is the discount factor, and $p_0 \in \Delta^S$ is the initial state distribution with $\Delta^k$ denoting the probability simplex in $k$-dimensions.

A policy is denoted by $\pi : \mathcal{S} \to \Delta^A$. When learning from demonstrations, we denote the expert's policy by $\pi_E : \mathcal{S} \to \mathcal{A}$. The rewards received by a policy at each state are $\boldsymbol{r}_\pi$ where $\boldsymbol{r}_\pi(s) = \mathbb{E}_{a \sim \pi(s)}[r(s, a)]$ and the transition probabilities for a policy $\pi$: $\boldsymbol{P}_\pi$, treated as a matrix, are defined as: $\boldsymbol{P}_\pi(s, s') = \mathbb{E}_{a \sim \pi(s)}[P(s, a, s')] = \sum_a \pi(a \mid s) P(s, a, s')$. We denote the state-action occupancies of policy $\pi$ as $\boldsymbol{u}_\pi \in \mathbb{R}^{S \cdot A}$, where $\boldsymbol{u}_\pi = (\boldsymbol{u}_\pi^{a_1\mathsf{T}}, \ldots, \boldsymbol{u}_\pi^{a_A\mathsf{T}})^\mathsf{T}$ and

$\boldsymbol{u}_\pi^a(s) = \mathbb{E}[\sum_{t=0}^{\infty} \gamma^t \cdot \mathbf{1}_{(s_t=s \wedge a_t=a)}]$. If we denote the reward function as a vector $\boldsymbol{r} \in \mathbb{R}^{S \cdot A}$, with $\boldsymbol{r} = (r(s_1, a_1), r(s_2, a_1), \dots, r(s_S, a_1), r(s_1, a_2), \dots, r(s_S, a_A))^\top$, then the expected return of policy $\pi$ under the reward function $r$ is denoted by $\rho(\pi, r) = \boldsymbol{u}_\pi^\top \boldsymbol{r}$.

**Linear Reward Functions** We assume, without loss of generality, that the reward function $\boldsymbol{r} \in \mathbb{R}^{S \cdot A}$ can be approximated as a linear combination of $k$ features $\boldsymbol{r} = \boldsymbol{\Phi} \boldsymbol{w}$, where $\boldsymbol{\Phi} \in \mathbb{R}^{S \cdot A \times k}$ is the linear feature matrix with rows as states and columns as features and $\boldsymbol{w} \in \mathbb{R}^k$. If $\boldsymbol{\Phi}$ is the identity matrix, then each state-action pair is allowed a unique reward. However, it is often the case that the rewards at different states are correlated via observable features which can be encoded in $\boldsymbol{\Phi}$. Note that the assumption of a linear reward function is not necessarily restrictive as these features can be arbitrarily complex nonlinear functions of the state and could be obtained via unsupervised learning from raw state observations [12, 13, 55]. Given $\boldsymbol{r} = \boldsymbol{\Phi} \boldsymbol{w}$, we denote the expected discounted feature counts of a policy as $\boldsymbol{\mu}_\pi = \boldsymbol{\Phi}^\top \boldsymbol{u}_\pi$, where $\boldsymbol{\mu}_\pi \in \mathbb{R}^k$. In this case, the return of a policy is given by $\rho(\pi, r) = \boldsymbol{u}_\pi^\top \boldsymbol{\Phi} \boldsymbol{w} = \boldsymbol{\mu}_\pi^\top \boldsymbol{w}$.

**Distributions over Reward Functions** We are interested in problems where there is uncertainty over the true reward function $r$. We will model this uncertainty as a distribution over $R$, the random variable representing the true reward function. This distribution could be a prior distribution $\mathbb{P}(R)$ that the agent has learned from previous tasks [63]. Alternatively the distribution could be the posterior distribution $\mathbb{P}(R \,|\, D)$ learned via Bayesian inverse reinforcement learning [46] given demonstrations $D$ or the posterior distribution $\mathbb{P}(R \,|\, R')$ learned via inverse reward design given a human-specified proxy reward function $R'$ [25]. While the distribution over $R$ may have an analytic form, this distribution is typically only available via sampling techniques such as Markov chain Monte Carlo (MCMC) sampling [11, 25, 46]. When there are no good priors for $R$, one may resort to Bayesian modeling techniques that mitigate the negative impacts of misspecified priors (e.g., [7]).

## 3.2 Risk Measures

**Value at Risk** When dealing with measures of risk, we assume that lower values are worse. Thus, as depicted in Figure 1, we want to maximize the value at risk (VaR) or conditional value at risk (CVaR). Given a risk-aversion parameter $\alpha \in [0, 1]$, the $\text{VaR}_\alpha$ is the $(1 - \alpha)$-quantile worst-case outcome. Thus, $\text{VaR}_\alpha$ can be written as $\text{VaR}_\alpha[X] = \sup\{x : \mathbb{P}(X \geq x) \geq \alpha\}$. Typical values of $\alpha$ for risk-sensitive applications are $\alpha \in [0.9, 1]$.

Despite the popularity of VaR, optimizing a policy for VaR has several problems: (1) VaR is not convex and leads to an NP hard optimization problem [16], (2) VaR ignores risk in the tail that occurs with probability less than $(1 - \alpha)$ which is problematic for domains where there are rare but catastrophic outcomes, and (3) VaR is not a coherent measure [3].

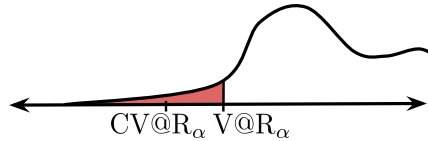

Figure 1: $\text{VaR}_\alpha$ measures the $(1 - \alpha)$-quantile worst-case outcome in a distribution. $\text{CVaR}_\alpha$ measures the expectation given that we only consider values less than the $\text{VaR}_\alpha$.

**Conditional Value at Risk** CVaR is a coherent risk measure [17] that is also commonly referred to as average value at risk, expected tail risk, or expected shortfall. For continuous atomless distributions, the CVaR is defined as

$$\text{CVaR}_\alpha[X] = \mathbb{E}[X \mid X \leq \text{VaR}_\alpha[X]]. \tag{1}$$

In addition to being coherent, CVaR is convex, and is a lower bound on VaR. CVaR is often preferable over VaR because it does not ignore the tail of the distribution and it is convex [48].

## 4 Balancing Risk and Return for Safe Imitation Learning

Let $\Pi$ be the set of all randomized policies, and let $\mathcal{R}$ be the set of all reward functions. Given some function $\psi : \Pi \times \mathcal{R} \to \mathbb{R}$ representing any performance metric for a policy under the unknown reward function $R \sim \mathbb{P}(R)$, we seek to find the policy that is the solution to the following problem:

$$\max_{\pi \in \Pi} \text{CVaR}_\alpha[\psi(\pi, R)] \tag{2}$$

The obvious choice for the performance metric is $\psi(\pi, r) = \rho(\pi, r)$. We discuss other choices in Section 4.2. We now discuss how to solve for the policy that optimizes Equation (2). We build on the classic LP formulation of MDP planning, which optimizes the state occupancy distribution subject to the Bellman flow constraints [45]. Specifically, we make use of the one-to-one correspondence between randomized policies $\pi : \mathcal{S} \to \Delta^A$ (where $A$ is the number of actions) and the state-action occupancy frequencies $\boldsymbol{u}_\pi$ [45]. This allows us to write $\max_\pi \rho(\pi, r)$ as the following linear program [45, 57]:

$$\max_{\boldsymbol{u} \in \mathbb{R}^{SA}} \left\{ \boldsymbol{r}^\mathsf{T} \boldsymbol{u} \mid \sum_{a \in \mathcal{A}} (\boldsymbol{I} - \gamma \cdot \boldsymbol{P}_a^\mathsf{T}) \boldsymbol{u}^a = \boldsymbol{p}_0, \boldsymbol{u} \geq \boldsymbol{0} \right\} . \tag{3}$$

We denote the posterior distribution over samples from $\mathbb{P}(R \mid D)$ as the vector $\boldsymbol{p}_R$, where each element of $\boldsymbol{p}_R$ represents the probability mass of one of the samples from the posterior distribution, e.g., $\boldsymbol{p}_R[i] = 1/N$ for $N$ sampled reward functions $R_1, R_2, R_3, \dots R_N$ obtained via MCMC [11, 46]. Because posterior distributions obtained via Bayesian IRL are usually discrete [11, 25, 46, 51], we cannot directly optimize for CVaR using the definition in (1) since this definition only works for atomless distributions (i.e. most continuous distributions). Instead, we use the following convex definition of CVaR [48] that works for any distribution (discrete or continuous):

$$\mathrm{CVaR}_\alpha[X] = \max_{\sigma \in \mathbb{R}} \left( \sigma - \frac{1}{1 - \alpha} \mathbb{E}[(\sigma - X)_+] \right) , \tag{4}$$

where $(x)_+ = \max(0, x)$ and the optimal $\sigma$ is equal to $\mathrm{VaR}_\alpha$ for atomless distributions [48]. Although we focus on the CVaR measure, our approach readily extends to other convex risk measures, such as the entropic risk [54]. The only difference is that our linear programs turn to tractable convex optimizations.

Writing the convex definition of CVaR in terms of a the probability mass vector $\boldsymbol{p}_R \in \mathbb{R}^N$, results in the following definition of the CVaR of a policy $\pi$ under the performance metric $\psi : \Pi \times \mathcal{R} \to \mathbb{R}$ and reward function random variable $R$:

$$\mathrm{CVaR}_\alpha[\psi(\pi, R)] = \max_{\sigma \in \mathbb{R}} \left( \sigma - \frac{1}{1 - \alpha} \mathbb{E}\big[ \big( \sigma - \psi(\pi, R) \big) \big]_+ \right) \tag{5}$$

$$= \max_{\sigma \in \mathbb{R}} \left( \sigma - \frac{1}{1 - \alpha} \boldsymbol{p}_R^\mathsf{T} [\sigma \cdot \boldsymbol{1} - \boldsymbol{\psi}(\pi, R)]_+ \right) , \tag{6}$$

where the boldface $\boldsymbol{\psi}(\pi, R) = \big( \psi(\pi, R_1), \dots, \psi(\pi, R_N) \big)^\mathsf{T}$ and $[\cdot]_+$ denotes the element-wise non-negative part of a vector: $[\boldsymbol{y}]_+ = \max\{\boldsymbol{y}, \boldsymbol{0}\}$. When the posterior distribution over $R$ is continuous, Equation (6) represents the Sample Average Approximation (SAA) method applied to (5), which is used extensively in stochastic programming [54] with known finite-sample properties [8]. One of the main insights of this chapter is that, using the same approach as the linear program above, we can formulate (2) as the following linear program which can be solved in polynomial time:

$$\max_{\boldsymbol{u} \in \mathbb{R}^{SA}, \sigma \in \mathbb{R}} \left\{ \sigma - \frac{1}{1 - \alpha} \boldsymbol{p}_R^\mathsf{T} [\sigma \cdot \boldsymbol{1} - \boldsymbol{\psi}(\pi, R)]_+ \mid \sum_{a \in \mathcal{A}} (\boldsymbol{I} - \gamma \cdot \boldsymbol{P}_a^\mathsf{T}) \boldsymbol{u}^a = \boldsymbol{p}_0, \boldsymbol{u} \geq \boldsymbol{0} \right\} . \tag{7}$$

Given the state-action occupancies $\boldsymbol{u}$ that maximize the above objective, the optimal policy can be recovered by appropriately normalizing these occupancies [45]. Thus, the optimal risk-averse IRL policy $\pi^\star$ can be constructed from an optimal $\boldsymbol{u}^\star$ solution to (7) as:

$$\pi^\star(s, a) = \frac{\boldsymbol{u}^\star(s, a)}{\sum_{a' \in \mathcal{A}} \boldsymbol{u}^\star(s, a')} . \tag{8}$$

## 4.1 Balancing Robustness and Expected Return

The above formulation in (7) finds a policy that has maximum CVaR. While this makes sense for highly risk-sensitive domains such as autonomous driving [51, 61] or medicine [4, 31], in other domains such as a robot vacuuming office carpets, we may also be interested in efficiency and performance, rather than pure risk-aversion. Even in highly risky situations, completely ignoring expected return and optimizing only for low probability events can lead to nonsensical behaviors that are overly cautious, such as an autonomous car deciding to never merge onto a busy highway [39].

To tune the risk-sensitivity of the optimized policy, we seek to solve for the policy that optimally balances performance and epistemic risk over the reward function. We formalize our goal via the parameter $\lambda \in [0, 1]$ and seek the policy that is the maximizer of the following optimization problem:

$$\max_{\pi \in \Pi} \quad \lambda \cdot \mathbb{E}[\psi(\pi, R)] + (1 - \lambda) \cdot \text{CVaR}_\alpha[\psi(\pi, R)] . \tag{9}$$

When $\lambda = 0$ we recover the fully robust policy, when $\lambda \in (0, 1)$ we obtain soft-robustness, and when $\lambda = 1$ we recover the risk-neutral Bayesian optimal policy [46]. We refer to the generalized problem in Equation (9) as *Bayesian Robust Optimization for Imitation Learning* or BROIL. Finally, by reformulating the optimization problem in Equation (7), we formulate BROIL as the following linear program:

$$\begin{aligned}
\underset{\boldsymbol{u} \in \mathbb{R}^{SA}, \sigma \in \mathbb{R}}{\text{maximize}} \quad & \lambda \cdot \boldsymbol{p}_R^\mathsf{T} \boldsymbol{\psi}(\pi_{\boldsymbol{u}}, R) + (1 - \lambda) \cdot \left( \sigma - \frac{1}{1 - \alpha} \boldsymbol{p}_R^\mathsf{T} [\sigma \cdot \boldsymbol{1} - \boldsymbol{\psi}(\pi_{\boldsymbol{u}}, R)]_+ \right) \\
\text{subject to} \quad & \sum_{a \in \mathcal{A}} \left( \boldsymbol{I} - \gamma \cdot \boldsymbol{P}_a^\mathsf{T} \right) \boldsymbol{u}^a = \boldsymbol{p}_0, \quad \boldsymbol{u} \geq \boldsymbol{0} ,
\end{aligned} \tag{10}$$

where we denote the stochastic policy that corresponds to a state-action occupancy vector $\boldsymbol{u}$ as $\pi_{\boldsymbol{u}}$.

## 4.2 Measures of Robustness

BROIL provides a general framework for optimizing policies that trade-off risk and return based on the specific choice of random variable $\psi(\pi, R)$, representing the desired measure of the safety or performance of a policy. We next describe two natural choices for defining $\psi(\pi, R)$.

**Robust Objective** If we seek a policy that is robust over the distribution $\mathbb{P}(R)$, we should optimize CVaR with respect to $\psi(\pi, R) = \rho(\pi, R)$, the expected return of the policy. Note that $R$ is a random variable so $\rho(\pi, R)$ is also a random variable that depends on the posterior distribution over $R$ and on $\pi$. In terms of the linear program (10) above we have $\boldsymbol{\psi}(\pi_{\boldsymbol{u}}, R) = \boldsymbol{R}^\mathsf{T} \boldsymbol{u}$, where $\boldsymbol{R}$ is a matrix of size $(S \cdot A) \times N$ where each column of $\boldsymbol{R}$ represents one sample of the vector over rewards for each state and action pair.

**Robust Baseline Regret Objective** If we have a baseline such as an expert policy or demonstrated trajectories, we may want maximize CVaR with respect to $\psi(\pi, R) = \rho(\pi, R) - \rho(\pi_E, R)$. This form of BROIL seeks to maximize the margin between the performance of the policy and the performance of the demonstrator. Rather than seeking to match the risk of the demonstrator [33], the Baseline Regret form of BROIL baselines its performance with respect to the random variable $\rho(\pi_E, R)$, while still trying to minimize tail risk. In terms of the linear program (10) above we have $\boldsymbol{\psi}(\pi_{\boldsymbol{u}}, R) = \boldsymbol{R}^\mathsf{T}(\boldsymbol{u} - \boldsymbol{u}_E)$. In practice, we typically only have samples of expert behavior rather than a full policy. In this case, we compute the empirical expected feature counts using a set of demonstrated trajectories $D = \{\tau_1, \ldots, \tau_m\}$ to get $\hat{\boldsymbol{\mu}}_E = \frac{1}{|D|} \sum_{\tau \in D} \sum_{(s_t, a_t) \in \tau} \gamma^t \phi(s_t, a_t)$, where $\phi : \mathcal{S} \times \mathcal{A} \to \mathbb{R}^k$ denotes the reward features. We then solve the above linear program (10) with with the performance metric $\boldsymbol{\psi}(\pi_{\boldsymbol{u}}, R) = \boldsymbol{R}^\mathsf{T} \boldsymbol{u} - \boldsymbol{W}^\mathsf{T} \hat{\boldsymbol{\mu}}_E$, where $\boldsymbol{W}$ is a matrix of size $k$-by-$N$ where each column $\boldsymbol{w}_i \in \mathbb{R}^k, i = 1, \ldots, N$ is a feature weight vector corresponding to each linear reward function $R_i$ sampled from the posterior such that $R_i = \boldsymbol{\Phi} \boldsymbol{w}_i$.

## 5 Experiments

In the next two sections we explore two case studies that highlight the performance and benefits of using BROIL for robust policy optimization. For the sake of interpretability, we keep the case studies simple; however, BROIL easily scales to much larger problems due to the efficiency of linear programming solvers. In the Appendix we empirically study the runtime of BIRL and demonstrate that BROIL can efficiently solve problems involving thousands of states in only a few hundred seconds of compute time on a personal laptop.[2]

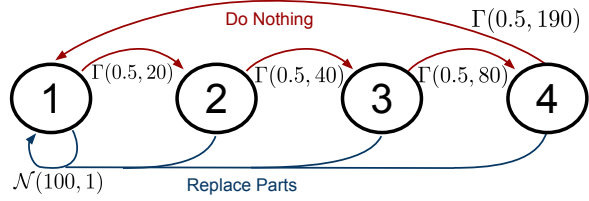

Figure 2: Machine Replacement MDP

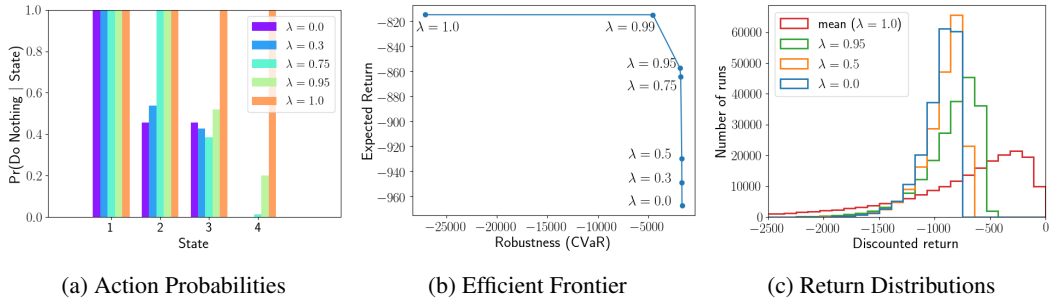

| (a) Action Probabilities | (b) Efficient Frontier | (c) Return Distributions |

Figure 3: Risk-sensitive ($\lambda \in [0, 1)$) and risk-neutral ($\lambda = 1$) policies for the machine replacement problem. Varying $\lambda$ results in a family of solutions that trade-off conditional value at risk and return. The risk-neutral policy has heavy tails, while BROIL produces risk-sensitive policies that trade-off a small decrease in expected return for a large increase in robustness (CVaR).

## 5.1 Zero-shot Robust Policy Optimization

We first consider the case where an agent wants to optimize a robust policy with respect to a prior over reward functions without access to expert demonstrations. This prior could come from historical data or from meta-learning on similar tasks [63].

We consider the machine replacement problem, a common problem in the robust MDP literature [16]. In this problem, there is a factory with a large number of machines with parts that are expensive to replace. There is also a cost associated with letting a machine age without replacing parts as this may cause damage to the machine, but this cost is uncertain. We model this problem as the MDP shown in Figure 2 with 4 states that represent the normal aging process of the machine, two actions in each state (replace parts or do nothing), discount factor $\gamma = 0.95$, and uniform initial state distribution. The prior distribution over the cost of the Do Nothing action is modeled as a gamma distribution $\Gamma(x, \theta)$, resulting in low expected costs but increasingly large tails as the machine ages. The prior distribution over the cost of replacing a part is modeled using a normal distribution.

Because we have no demonstrations, we use the Robust Objective version of BROIL (Section 4.2). We sampled 2000 reward functions from the prior distributions over costs and computed the CVaR optimal policy with $\alpha = 0.99$ for different values of $\lambda$. Figure 3(a) shows the action probabilities of the optimal policy under different values of $\lambda$, where $\mathbb{P}(\text{Replace Parts}) = 1 - \mathbb{P}(\text{Do Nothing})$. Setting $\lambda = 1$ gives the optimal policy with respect to the mean reward under the reward posterior. This policy is risk-neutral and chooses to never repair the machine since the mean of the gamma distribution is $x \cdot \theta$, so in expectation it is optimal to do nothing. As $\lambda$ decreases, the optimal policy hedges more against tail risk via a stochastic policy that sometimes repairs the machine. With $\lambda = 0$, we recover the robust optimal policy that only seeks to optimize CVaR. This policy is maximally risk-sensitive and chooses to probabilistically repair the machine in states 2 and 3 and always repair in state 4 to avoid the risk of doing nothing and incurring a possibly high cost. Figure 3(b) shows the efficient frontier of Pareto optimal solutions. BROIL achieves significant improvements in robustness by sacrificing a small amount of expected utility. Figure 3(c) shows that the BROIL policies with $\lambda < 1$ have much smaller tails than the policy that only optimizes with respect to the expected rewards.

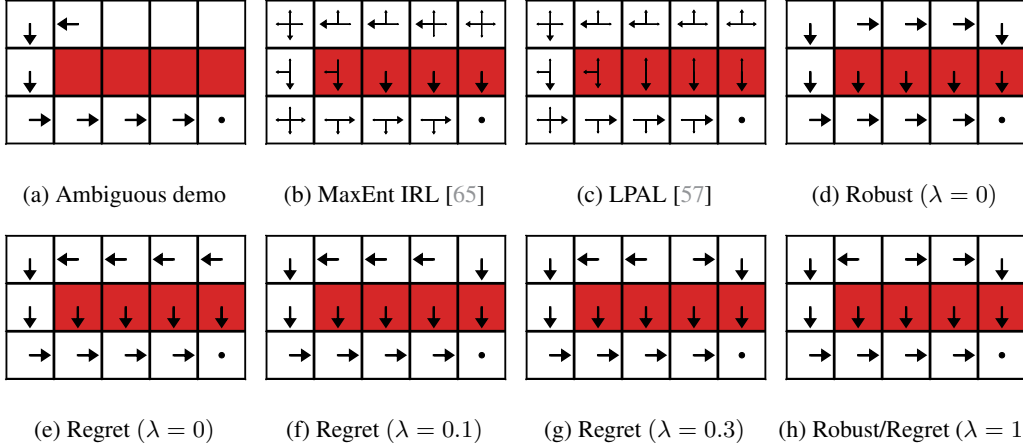

| (a) Ambiguous demo | (b) MaxEnt IRL [65] | (c) LPAL [57] | (d) Robust ($\lambda = 0$) |
|---|---|---|---|
| (e) Regret ($\lambda = 0$) | (f) Regret ($\lambda = 0.1$) | (g) Regret ($\lambda = 0.3$) | (h) Robust/Regret ($\lambda = 1$) |

Figure 4: When demonstrations BROIL results in a family of solutions that balance return and risk based on the value of $\lambda$. (a) Ambiguous demonstration that does not convey enough information to determine how undesireable the red states are. (b-c) MaxEnt IRL and LPAL results in stochastic policies where size of arrow reprents probability. (d) The robust policy with $\lambda = 0$ balances the goodness and badness of red and prefers taking a shortcut. (e-g) The regret policy avoids red for small $\lambda$. (h) The optimal policy for the mean reward ($\lambda = 1$) takes a short cut through red cells.

## 5.2   Ambiguous Demonstrations

Next we consider the case where the agent has no prior knowledge about the reward function, but where demonstrations are available. In particular, we are interested in the case where demonstrations cover only part of the state-space, so even after observing a demonstration there is still high uncertainty over the true reward function. To clearly showcase the benefits of BROIL, we constructed the MDP shown in Figure 4 where there are two features (red and white) with unknown costs, a terminal state in the bottom right, and $\gamma = 0.95$. Actions are in the four cardinal directions with deterministic dynamics. The agent observes the demonstration shown in Figure 4(a) that demonstrates some preference for the white feature over the red feature and a preference for exiting the MDP. However, the demonstration does not provide sufficient information to know what to do in the top right states where demonstrator actions are unavailable. In particular, the agent does not know the true cost of the red cells and whether taking the shortest path from the top right states to the terminal state is optimal. We demonstrate that BROIL results in much more sensible policies across a spectrum of risk-sensitives, than other state-of-the-art approaches.

Given the single demonstration, we generated 2000 samples from the posterior $\mathbb{P}(R \mid D)$ using Bayesian IRL [46]. We compare against the risk-sensitive, maxmin algorithm, LPAL, proposed by Syed et al. [57] and the risk-neutral Maximum Entropy IRL algorithm [65]. Shown in Figure 4 are the optimal policies for MaxEnt IRL [65], LPAL [57], and BROIL using the robust and baseline regret formulations with $\alpha = 0.95$. We plotted the unique policies and a sample $\lambda$ that results in each policy. Note that $\lambda = 1$ is equivalent to solving for the optimal policy for the mean reward Figure 4(h). The baseline regret formulation uses the expert feature counts to baseline risk and seeks to completely avoid the red feature for $\lambda = 0$. As $\lambda$ increases, the baseline regret policy is more willing to take a shortcut to get to the terminal state in the bottom right corner. Conversely, the robust policy takes the shortcut through the far right red cell which balances the risk of the red feature with the knowledge that the white feature is likely to also have high cost. The reason the robust policy does not match the demonstration for one state in Figure 4(d) is that Bayesian IRL does not assume demonstrator optimality, only Boltzman rationality. We used a relatively small inverse temperature parameter ($\beta = 10$) resulting in reward function hypotheses that allow for occasional demonstrator errors. Using a large inverse temperature causes the robust policy to match all the demonstrator's actions (see the Appendix for more details regarding Bayesian IRL).

To better understand the differences between these approaches without committing to a particular ground-truth reward function, we examine each algorithm's performance across the posterior distribution $\mathbb{P}(R \mid D)$. Figure 5(a) shows $\psi(\pi, R) = \rho(\pi, R)$ sorted from smallest to largest

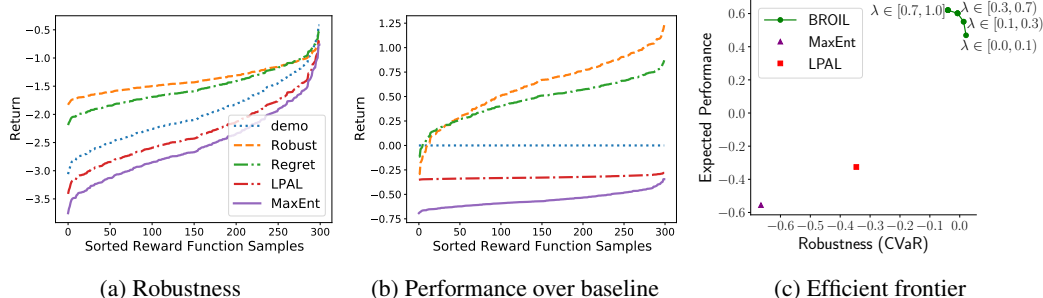

| (a) Robustness | (b) Performance over baseline | (c) Efficient frontier |

Figure 5: Sorted return distributions over the posterior for the BROIL Robust and Baseline Regret policies compared to the return distributions of the demonstration, MaxEnt IRL [65], LPAL [57]. The robust policy attempts to maximize worst-case performance over the posterior. The baseline regret also seeks to maximize worst-case performance but relative to the demonstration.

when evaluated under each sample from the posterior. Figure 5(b) shows the results when $\psi(\pi, R) = \rho(\pi, R) - \rho(\pi_E, R)$. LPAL is similar to the baseline regret formulation of BROIL in that it seeks to optimize a policy that performs better than the demonstrator; however, unlike BROIL, LPAL uses a fully adversarial maxmin approach that penalizes the biggest deviation from the demonstrated feature counts [57]. This results in always avoiding red cells, but also trying to exactly match the feature counts of the demonstration. This feature count matching results in a highly stochastic policy that does not always terminate quickly. MaxEnt IRL is completely risk-neutral, but also seeks to explicitly match feature counts while maintaining maximum entropy over the policy actions. This results in a highly stochastic policy that sometimes takes shortcuts through the red cells, but also sometimes takes actions that move it away from the terminal state.

Figure 5 shows that both formulations of BROIL significantly outperform MaxEnt IRL and LPAL. The return distribution of the robust BROIL policy is flatter than the other policies as it attempts to find a policy that performs well in the 5% worst-case under all reward functions and needs to be robust to posterior samples that put high costs on white cells and only slightly higher costs on red. On the other hand the Baseline Regret formulation computes risk under the posterior with respect to the expected feature counts $\hat{\mu}_E$ of the demonstrator. This makes reward function hypotheses that would lead to entering red states more risky since the demonstrator only visited white states. The regret formulation seeks to maximize the margin between the return of the baseline regret policy and the return of the demonstration over the posterior. Thus, the regret policy tracks the performance of the baseline more than the robust policy as shown in Figure 5(a). As shown in Figure 5(b), the regret formulation has better tail performance with respect to the posterior baseline regret. Figure 5(c) shows the efficient frontier for the baseline regret formulation and shows that BROIL dominates LPAL and MaxEnt IRL with respect to both expected return and robustness.

## 6 Conclusion and Future Work

We proposed Bayesian Robust Optimization for Imitation Learning (BROIL), a method for optimizing a policy to be robust to conditional value at risk under an unknown reward function. Our results show that BROIL has better overall performance than existing risk-sensitive maxmin [57] and risk-neutral [65] approaches to IRL. Our approach balances return and conditional value at risk to produce a family of robust solutions parameterized by the risk-aversion of the user. This work focuses on policy optimization and requires either a prior or posterior distribution over likely reward functions. However, obtaining a posterior via Bayesian IRL [46] typically involves repeatedly solving an MDP in the inner loop which makes it difficult to obtain posterior distributions in complex control tasks. Future work includes taking advantage of recent research on efficient non-linear Bayesian reward learning via Gaussian processes [9] and deep neural networks [12]. Future work also includes investigating natural extensions of our work to continuous state and action spaces such as optimizing the BROIL objective via policy gradient methods [53, 56] or approximate linear programming [19, 41, 42], applying BROIL to more complex domains such as health care and robotics, and investigating extensions to deep Bayesian inverse reinforcement learning [12], meta inverse reinforcement learning [63], and inverse reward design [25].

## Broader Impact

Algorithms that balance risk and return are have been common in financial applications for a long time, but are just starting to be applied to AI/ML systems. We believe this is a positive trend as many AI/ML applications have risk and return trade-offs that are not always adequately addressed. In this work we have proposed a principled approach optimizing control policies that balance expected return and epistemic risk under an uncertain reward functions. We see this work as an important step towards the general goal of robust autonomous systems that can interact safely with and assist humans in a wide variety of tasks and under a wide variety of preferences and risk tolerances. However, there are potential downsides to having risk and return trade-offs if these trade-offs are made incorrectly or interpreted incorrectly—despite using risk-sensitive metrics, financial systems still occasionally crash or fail. Our proposed algorithm, BROIL, does not guarantee safety, thus an autonomous system based on our approach will not be guaranteed to never make a mistake. Instead, BROIL optimizes a policy that is robust with respect to the agent's uncertainty over its learned representation of the demonstrator's reward function. Thus, the optimized policy may not always conform to a human's intuition about what safe or robust behavior should look like.

## Acknowledgments and Disclosure of Funding

We would like to thank the reviewers for their detailed feedback that helped to improve the paper. This work has taken place in the Personal Autonomous Robotics Lab (PeARL) at the University of Texas at Austin and the Reinforcement Learning and Robustness Lab (RLsquared) at the University of New Hampshire. PeARL research is supported in part by the NSF (IIS-1724157, IIS-1638107,IIS-1617639, IIS-1749204) and ONR(N00014-18-2243). This research was also sponsored by the Army Research Office and was accomplished under Cooperative Agreement Number W911NF-19-2-0333. RLsquared research is supported in part by NSF Grants IIS-1717368 and IIS-1815275. The views and conclusions contained in this document are those of the authors and should not be interpreted as representing the official policies, either expressed or implied, of the Army Research Office or the U.S. Government. The U.S. Government is authorized to reproduce and distribute reprints for Government purposes notwithstanding any copyright notation herein.

## Footnotes

\*Work done while at UT Austin.

[2]Code to reproduce experiments is available at `https://github.com/dsbrown1331/broil`

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
