[Supplementary Material]

# Supplementary Materials for
# Bayesian Robust Optimization for Imitation Learning

**Daniel S. Brown**[*]
UC Berkeley
dsbrown@berkeley.edu

**Scott Niekum**
University of Texas at Austin
sniekum@cs.utexas.edu

**Marek Petrik**
University of New Hampshire
mpetrik@cs.unh.edu

## A  Code

Code to reproduce all experiments is available at `https://github.com/dsbrown1331/broil`.

## B  Linear Programming Details

The soft-robust BROIL objective is:

$$\underset{\boldsymbol{u}\in\mathbb{R}^{SA},\,\sigma\in\mathbb{R}}{\text{maximize}} \quad \lambda\cdot\boldsymbol{p}^{\mathsf{T}}\boldsymbol{\psi}(\pi_{\boldsymbol{u}},R) + (1-\lambda)\cdot\left(\sigma - \frac{1}{1-\alpha}\boldsymbol{p}^{\mathsf{T}}\left[\sigma\cdot\boldsymbol{1} - \boldsymbol{\psi}(\pi_{\boldsymbol{u}},R)\right]_{+}\right)$$
$$\text{subject to} \quad \sum_{a\in\mathcal{A}}\left(\boldsymbol{I} - \gamma\cdot\boldsymbol{P}_a^{\mathsf{T}}\right)\boldsymbol{u}^a = \boldsymbol{p}_0, \quad \boldsymbol{u}\geq\boldsymbol{0}\,.$$

When using the robust performance metric described in Section 4.2, we have $\boldsymbol{\psi}(\pi_{\boldsymbol{u}},R) = \boldsymbol{R}^{\mathsf{T}}\boldsymbol{u}$, where $\boldsymbol{R}$ is a matrix of size $(S\cdot A)\times N$ where each column of $\boldsymbol{R}$ represents one sample of the vector over rewards for each state and action pair. This results in the following optimization problem:

$$\underset{\boldsymbol{u}\in\mathbb{R}^{SA},\,\sigma\in\mathbb{R}}{\text{maximize}} \quad \lambda\cdot(\boldsymbol{R}\boldsymbol{p})^{\mathsf{T}}\boldsymbol{u} + (1-\lambda)\cdot\left(\sigma - \frac{1}{1-\alpha}\boldsymbol{p}^{\mathsf{T}}\left[\sigma\cdot\boldsymbol{1} - \boldsymbol{R}^{\mathsf{T}}\boldsymbol{u}\right]_{+}\right)$$
$$\text{subject to} \quad \sum_{a\in\mathcal{A}}\left(\boldsymbol{I} - \gamma\cdot\boldsymbol{P}_a^{\mathsf{T}}\right)\boldsymbol{u}^a = \boldsymbol{p}_0, \quad \boldsymbol{u}\geq\boldsymbol{0}\,.$$

where $\boldsymbol{R}\boldsymbol{p}$ is the mean reward under the posterior distribution.

This can be written as a linear program in standard form as:

$$-\underset{\boldsymbol{u}\in\mathbb{R}^{SA},\,\boldsymbol{z}\in\mathbb{R}^{N},\,\sigma\in\mathbb{R}}{\text{minimize}} \quad -\lambda\cdot\boldsymbol{p}^{\mathsf{T}}\boldsymbol{R}^{\mathsf{T}}\boldsymbol{u} - (1-\lambda)\cdot\left(\sigma + \frac{1}{1-\alpha}\boldsymbol{p}^{\mathsf{T}}\boldsymbol{z}\right)$$
$$\text{subject to} \quad \sigma\cdot\boldsymbol{1} - \boldsymbol{R}^{\mathsf{T}}\boldsymbol{u} - \boldsymbol{z} \leq 0,$$
$$\left[(\boldsymbol{I}-\gamma\boldsymbol{P}_{a_1}^{\mathsf{T}}),\ldots,(\boldsymbol{I}-\gamma\boldsymbol{P}_{a_m}^{\mathsf{T}})\right]\begin{bmatrix}\boldsymbol{u}^{a_1}\\ \vdots \\ \boldsymbol{u}^{a_n}\end{bmatrix} = \boldsymbol{p}_0,$$
$$\boldsymbol{u}\geq\boldsymbol{0},\ \boldsymbol{z}\geq\boldsymbol{0}\,.$$

We solve the above linear program to obtain the results presented in Section 5.1.

---

[*]Work done while at UT Austin.

When using the baseline regret performance metric, $\boldsymbol{\psi}(\pi_{\boldsymbol{u}}, R) = \boldsymbol{R}^{\mathsf{T}}(\boldsymbol{u} - \boldsymbol{u}_E)$, we have the following optimization problem:

$$\underset{\boldsymbol{u} \in \mathbb{R}^{SA},\, \sigma \in \mathbb{R}}{\text{maximize}} \quad \lambda \cdot (\boldsymbol{R}\boldsymbol{p})^{\mathsf{T}}(\boldsymbol{u} - \boldsymbol{u}_E) + (1 - \lambda) \cdot \left(\sigma - \frac{1}{1 - \alpha}\boldsymbol{p}^{\mathsf{T}}\left[\sigma \cdot \mathbf{1} - \boldsymbol{R}^{\mathsf{T}}(\boldsymbol{u} - \boldsymbol{u}_E)\right]_+\right)$$

$$\text{subject to} \quad \sum_{a \in \mathcal{A}}\left(\boldsymbol{I} - \gamma \cdot \boldsymbol{P}_a^{\mathsf{T}}\right)\boldsymbol{u}^a = \boldsymbol{p}_0, \quad \boldsymbol{u} \geq \mathbf{0},$$

This can be written as a linear program in standard form as follows:

$$-\underset{\boldsymbol{u} \in \mathbb{R}^{SA},\, \boldsymbol{z} \in \mathbb{R}^{N},\, \sigma \in \mathbb{R}}{\text{minimize}} \quad -\lambda \cdot \boldsymbol{p}^{\mathsf{T}}\boldsymbol{R}^{\mathsf{T}}(\boldsymbol{u} - \boldsymbol{u}_E) - (1 - \lambda) \cdot (\sigma + \frac{1}{1 - \alpha}\boldsymbol{p}^{\mathsf{T}}\boldsymbol{z})$$

$$\text{subject to} \quad \sigma \cdot \mathbf{1} - \boldsymbol{R}^{\mathsf{T}}\boldsymbol{u} - \boldsymbol{z} \leq -\boldsymbol{R}^{\mathsf{T}}\boldsymbol{u}_E$$

$$\left[(\boldsymbol{I} - \gamma\boldsymbol{P}_{a_1}^{\mathsf{T}}), \ldots, (\boldsymbol{I} - \gamma\boldsymbol{P}_{a_m}^{\mathsf{T}})\right] \begin{bmatrix} \boldsymbol{u}^{a_1} \\ \vdots \\ \boldsymbol{u}^{a_n} \end{bmatrix} = \boldsymbol{p}_0$$

$$\boldsymbol{u} \geq \mathbf{0},\ \boldsymbol{z} \geq \mathbf{0}.$$

Typically, we only have access to a handful of demonstrations and do not have direct access to the state-occupancies of the demonstrator and cannot accurately estimate them. If we assume the reward function is a linear combination of features, it is often the case that the number of features $k$ is much less than the total number of state-action pairs. Thus, it is typically much more practical and computationally accurate to use an empirical estimate of the expert's expected feature counts and rewrite the baseline regret as $\boldsymbol{\psi}(\pi_{\boldsymbol{u}}, R) = \boldsymbol{R}^{\mathsf{T}}\boldsymbol{u} - \boldsymbol{W}^{\mathsf{T}}\hat{\boldsymbol{\mu}}_E$, where $\boldsymbol{W}$ is a matrix of size $k$-by-$N$ where each column is a feature weight vector $\boldsymbol{w} \in \mathbb{R}^k$ corresponding to each linear reward function weight vector sampled from the posterior as described in Section 4.2. This results in the following linear program which we use for our experiments in Section 5.2.

$$-\underset{\boldsymbol{u} \in \mathbb{R}^{SA},\, \boldsymbol{z} \in \mathbb{R}^{N},\, \sigma \in \mathbb{R}}{\text{minimize}} \quad -\lambda \cdot \boldsymbol{p}^{\mathsf{T}}(\boldsymbol{R}^{\mathsf{T}}\boldsymbol{u} - \boldsymbol{W}^{\mathsf{T}}\hat{\boldsymbol{\mu}}_E) - (1 - \lambda) \cdot (\sigma + \frac{1}{1 - \alpha}\boldsymbol{p}^{\mathsf{T}}\boldsymbol{z})$$

$$\text{subject to} \quad \sigma \cdot \mathbf{1} - \boldsymbol{R}^{\mathsf{T}}\boldsymbol{u} - \boldsymbol{z} \leq -\boldsymbol{W}^{\mathsf{T}}\hat{\boldsymbol{\mu}}_E$$

$$\left[(\boldsymbol{I} - \gamma\boldsymbol{P}_{a_1}^{\mathsf{T}}), \ldots, (\boldsymbol{I} - \gamma\boldsymbol{P}_{a_m}^{\mathsf{T}})\right] \begin{bmatrix} \boldsymbol{u}^{a_1} \\ \vdots \\ \boldsymbol{u}^{a_n} \end{bmatrix} = \boldsymbol{p}_0$$

$$\boldsymbol{u} \geq \mathbf{0},\ \boldsymbol{z} \geq \mathbf{0}.$$

We use Scipy's linear programming software (v 1.4.1) when solving the above linear programs in the experiments in the paper.[2] Note also that the term $-\lambda\boldsymbol{p}^{\mathsf{T}}\boldsymbol{R}^{\mathsf{T}}\boldsymbol{u}_E$ or $-\lambda\boldsymbol{p}^{\mathsf{T}}\boldsymbol{W}^{\mathsf{T}}\hat{\boldsymbol{\mu}}_E$ in the baseline regret linear program objectives above can be dropped since they are just constants that do not affect the resulting optimal policies.

## C  Runtime and Scalability

In the main paper we focused on simple case studies that are easily interpretable; however, our method readily scales to much larger problems. In Figure 1 we show that we can easily solve instances with thousands of states and thousands of reward functions in the posterior. To further test the runtime of BROIL, we optimized a robust policy for a 60-by-60 gridworld (3,600 states) which took an average time of 119.34 seconds to solve the BROIL linear program given a reward function distribution. For comparison, a CVaR optimization approach for MDPs with no uncertainty over the reward function takes 2 hours for a similar-sized gridworld (see [3] Section 5, last paragraph). All experiments were run using Scipy's standard linear programming solver on a Dell Inspiron 5577 laptop with an Intel i7-7700 processor.

Figure 1: LP runtime as number of states and number of reward function hypothesis are increased for the machine replacement problem. Results are averaged over 20 trials and error bars show plus or minus one standard deviation. (a) Runtime as the number of states is increased and number of reward function hypotheses is fixed at 200. (b) Runtime as the number of reward function hypotheses is increased and the number of states is fixed at 100.

## D  Bayesian IRL Details

When learning a posterior from demonstrations we use Bayesian IRL [4]. Bayesian IRL has the following likelihood function: Bayesian IRL assumes access to an MDP without a reward function, denoted MDP\R and a set of demonstrations, $D = \{(s_1, a_1), \ldots, (s_m, a_m)\}$, consisting of state-action pairs. Bayesian IRL (BIRL) [4] seeks to estimate the posterior over reward functions given demonstrations, $\mathbb{P}(R \mid D) \propto \mathbb{P}(D \mid R) \cdot \mathbb{P}(R)$. BIRL makes the assumption that the demonstrator is Boltzmann rational and follows a soft-max policy, resulting in the likelihood function

$$\mathbb{P}(D \mid R) = \prod_{(s,a) \in D} \mathbb{P}\big((s,a) \mid R\big) = \prod_{(s,a) \in D} \frac{e^{\beta Q_R^*(s,a)}}{\sum_{b \in A} e^{\beta Q_R^*(s,b)}} \tag{1}$$

where $Q_R^*(s, a)$ is the optimal Q-value function for reward $R$, and $\beta$ is a parameter representing the confidence in the demonstrator's optimality. Given a reward function $R$, the Q-value of a state-action pair $(s, a)$ is defined as $Q_R^\pi(s, a) = R(s) + \gamma \sum_{s' \in S} T(s, a, s') V_R^\pi(s')$. We denote $Q_R^*(s, a) = \max_{\pi \in \Pi} Q_R^\pi(s, a)$. Equation 1 gives greater likelihood to rewards for which the actions taken by the expert have higher Q-values than the alternative actions.

Bayesian IRL uses Markov chain Monte Carlo (MCMC) sampling to sample from the posterior $\mathbb{P}(R \mid D)$. Feature weights are sampled according to a proposal distribution, and for each sample the MDP is solved to obtain the sample's likelihood and determine the transition probabilities within the Markov chain. We use $\beta = 10$ for all of our experiments. We sample reward function weights for MCMC by using a Gaussian proposal distribution centered arounnd the previous sample, with standard deviation of 0.2. We use a burn-in period of 500 samples and skip every 5th sample after that to reduce auto-correlation. We experimented with range of values for $\beta$ and found very similar results. The step size was tuned to result in an accept ratio close to 0.4. Because scaling a reward function does not affect the optimal policy, we following prior work [1, 2, 5] and assume that the reward function is scaled. We project each sampled reward function weight proposal to the $L_2$-norm ball to ensure that $\|w\|_2 = 1$.

## E  Maximum Entropy IRL Detais

We compare against Maximum Entropy IRL [6]. We use the implementation presented by Ziebart et al. [6], but to make it more comparable to Bayesian IRL we also add a Boltzmann parameter $\beta$ to the likelihood such that

$$\mathbb{P}(\xi) \propto \exp(\beta R(\xi)). \tag{2}$$

where $\xi$ is a trajectory and $R(\xi)$ is the cumulative return of a trajectory. We use $\beta = 10$ to match our implementation choice for Bayesian IRL. We used a learning rate of 0.01. We perform projected gradient descent by projecting to the $L_2$-norm ball such that $\|w\|_2 = 1$. We use a horizon equal to the number of states in the MDP. We stop gradient ascent on the likelihood function once it has converged. We detect convergence by measuring the difference in the $L_2$-norm of the updated and prior weights and check if that is within a precision value of $0.00001$. If so, then we stop gradient ascent. We experimented with several different values for each of these hyperparameters and found these to provide best performance.

## F  LPAL Details

Linear Programming Apprenticeship Learning (LPAL) [5] has a robust form that is similar to ours but makes several critical and limiting assumptions: (1) they assume that the reward weights are strictly positive, this means they assume that the feature vector $\phi(s)$ explicitly encodes whether a feature is good or bad by its sign. (2) They assume very accurate estimation of the expert's expected feature counts $\hat{\mu}_E$. This requires an extremely large number of demonstrations (c.f. [2]). (3) Finally, they assume a worst-case adversarial reward function that penalizes whatever the learner does that is most different from the demonstrator, even if this reward function completely contradicts the demonstrations, i.e., it does not take into account the likelihood of reward functions.

To compare BROIL against a state-of-the-art robust IRL approach, we implemented Linear Programming Apprenticeship Learning [5]. The original paper assumes that the signs of the feature weights determine whether a feature is good or bad and that the feature weights $w$ lie on the probability simplex. In our work we do not assume prior knowledge about which features are good or bad (we seek to infer this from demonstrations). Thus, we implemented LPAL in a way that allows it to work with any features and feature weights that can be both positive and negative. We simply assume that $\|w\|_1 \leq 1$.

In the paper we compare against the solution to the following derivation of the LPAL algorithm which does not assume the weights are non-negative, thus removing the need to know beforehand which features are good or bad. The LPAL formulation we use is as follows:

$$- \min_{B \in \mathbb{R},\, \boldsymbol{u} \in \mathbb{R}^{S \cdot A}} \quad B \tag{3}$$

$$\text{s.t.} \quad B \cdot \mathbf{1} - \boldsymbol{\Phi}^{\mathsf{T}} \boldsymbol{u} \leq -\hat{\boldsymbol{\mu}}_E, \tag{4}$$

$$-B \cdot \mathbf{1} + \boldsymbol{\Phi}^{\mathsf{T}} \boldsymbol{u} \leq \hat{\boldsymbol{\mu}}_E, \tag{5}$$

$$\left[ (\boldsymbol{I} - \gamma \boldsymbol{P}_{a_1}^{\mathsf{T}}), \ldots, (\boldsymbol{I} - \gamma \boldsymbol{P}_{a_m}^{\mathsf{T}}) \right] \begin{bmatrix} \boldsymbol{u}_{a_1} \\ \vdots \\ \boldsymbol{u}_{a_n} \end{bmatrix} = \boldsymbol{p}_0, \tag{6}$$

$$\boldsymbol{u} \geq \mathbf{0}, \tag{7}$$

$$B \in \mathbb{R}, \tag{8}$$

where $\boldsymbol{\Phi} \in \mathbb{R}^{S \cdot A \times k}$ is the linear feature matrix with rows as states and columns as features and $\boldsymbol{w} \in \mathbb{R}^k$.

We now derive the above formulation of the maxmin objective for LPAL. The basic LPAL objective is

$$\max_{\boldsymbol{u} \in \mathcal{U}} \min_{\boldsymbol{w} \geq \mathbf{0}, \|\boldsymbol{w}\|_1 \leq 1} (\boldsymbol{u}^{\mathsf{T}} \boldsymbol{\Phi} \boldsymbol{w} - \boldsymbol{u}_E^{\mathsf{T}} \boldsymbol{\Phi} \boldsymbol{w}), \tag{9}$$

where $\mathcal{U}$ is the set of all feasible state-action occupancies.

If we want to get rid of the requirement for positive weights then we have

$$\max_{\boldsymbol{u} \in \mathcal{U}} \min_{\|\boldsymbol{w}\|_1 \leq 1} (\boldsymbol{u}^{\mathsf{T}} \boldsymbol{\Phi} \boldsymbol{w} - \boldsymbol{u}_E^{\mathsf{T}} \boldsymbol{\Phi} \boldsymbol{w}) \tag{10}$$

The inner minimization can be changed into a maximization as follows:

$$\max_{\boldsymbol{u} \in \mathcal{U}} - \max_{\|\boldsymbol{w}\|_1 \leq 1} (-\boldsymbol{\Phi}^{\mathsf{T}} \boldsymbol{u} + \boldsymbol{\Phi}^{\mathsf{T}} \boldsymbol{u}_E)^{\mathsf{T}} \boldsymbol{w} \tag{11}$$

Next, we use the fact that the $L_\infty$ norm and $L_1$ norm are dual to each other and that $\|\boldsymbol{z}\| = \|-\boldsymbol{z}\|$ to get the following optimization problem:

$$\max_{\boldsymbol{u}\in\mathcal{U}} \, -\|\boldsymbol{\Phi}^\mathsf{T}\boldsymbol{u} - \boldsymbol{\Phi}^\mathsf{T}\boldsymbol{u}_E\|_\infty \tag{12}$$

We change the maximization to a minimization by changing signs:

$$-\min_{\boldsymbol{u}\in\mathcal{U}} \, \|\boldsymbol{\Phi}^\mathsf{T}\boldsymbol{u} - \boldsymbol{\Phi}^\mathsf{T}\boldsymbol{u}_E\|_\infty \tag{13}$$

Using a standard linear programming reformulation we can write the above objective as follows:

$$-\min_{\boldsymbol{u}\in\mathcal{U},\, B\in\mathbb{R}} \, \left\{ B \,\middle|\, B\cdot\boldsymbol{1} \geq \boldsymbol{\Phi}^\mathsf{T}\boldsymbol{u} - \boldsymbol{\Phi}^\mathsf{T}\boldsymbol{u}_E, \; -B\cdot\boldsymbol{1} \geq -\boldsymbol{\Phi}^\mathsf{T}\boldsymbol{u} + \boldsymbol{\Phi}^\mathsf{T}\boldsymbol{u}_E \right\}. \tag{14}$$

Rather than assuming access to the state-action occupancies of the demonstrator, we will typically use a finite number of demonstrations to come up with an empirical estimate of the demonstrator's expected feature counts $\boldsymbol{\mu}_E = \boldsymbol{\Phi}^\mathsf{T}\boldsymbol{u}_E$. Given a set of demonstrated trajectories $D = \{\tau_1, \ldots, \tau_m\}$ we compute $\hat{\boldsymbol{\mu}}_E = \frac{1}{|D|}\sum_{\tau\in D}\sum_{(s_t,a_t)\in\tau}\gamma^t\phi(s_t,a_t)$, where $\phi : \mathcal{S}\times\mathcal{A} \to \mathbb{R}^k$ denotes the state-action reward features such that $r(s,a) = \boldsymbol{w}^\mathsf{T}\phi(s,a)$. This gives us the following linear program:

$$-\min_{\boldsymbol{u}\in\mathcal{U},\, B\in\mathbb{R}} \, \left\{ B \,\middle|\, B\cdot\boldsymbol{1} \geq \boldsymbol{\Phi}^\mathsf{T}\boldsymbol{u} - \hat{\boldsymbol{\mu}}_E, \; -B\cdot\boldsymbol{1} \geq -\boldsymbol{\Phi}^\mathsf{T}\boldsymbol{u} + \hat{\boldsymbol{\mu}}_E \right\}. \tag{15}$$

## Footnotes

[2]`https://docs.scipy.org/doc/scipy/reference/generated/scipy.optimize.linprog.html`