[Reviews · NeurIPS 2020]

Review 1

Summary and Contributions: The paper considers the problem of risk averse optimization with respect to a distribution of reward functions. They take advantage of the dual of the LP representation of value iteration in order to define a risk averse policy optimization method for discrete MDPs. They show how to use this to do risk averse optimization and imitation learning in example MDPs. Contributions: 1) a linear programming formulation to compute the optimal policy for CVaR; 2) show how to use this to implement robust policy optimization under a prior and robust imitation learning; 3) demonstrate favorable comparisons with existing risk-sensitive and risk neutral algorithms for both settings. Right now I think that the description of the contributions hides the most useful contribution.

Strengths: The paper considers an important problem: risk averse optimization of policies under a distribution of reward functions. The paper provides a mathematically elegant way to incorporate risk-aversion into a standard and moderately scalable MDP solution technique. To my knowledge, the proposed approach is novel, easy to implement, and will improve the state-of-the-art on an important problem for the NeurIPS community.

Weaknesses: While I believe that the proposed approach does scale well, I think the evaluation from the authors is somewhat incomplete. It would be nice to see this deployed in larger problems. The authors mention that they are somewhat limited by the size of problems that they can run inference in --- I think this is not true. It is possible to test this algorithm as a planning method in large problems, either with a hand-coded prior distribution over rewards or by transferring a learned reward function to a new environment (this is where this method should do quite well.) As it stands, the reader of this paper is left to guess at the scalability of this approach based on general knowledge about LP solvers.

Correctness: I believe the claims and the method are correct. I think that the empirical methodology is sound, but somewhat weak. I would like to see the authors demostrate that this method works for substantially larger planning problems. Even if it is only over larger problems with hand-specified distributions over reward functions. I have one concern, which is that the robust implementation of the method (4.d) doesn't match the demonstration for one state. I think this is because the planner is actually indifferent between two actions based on the inferred reward, but it would be good to see some discussion of this.

Clarity: Overall, I think the paper is fairly well written. I understand that the authors are working within the page restrictions of the conference. With that said, I think there is substantial room for improvement in the paper presentation. First, I think there are more specific ways to describe the contributions (copied from summary): 1) a linear programming formulation to compute the optimal policy for CVaR; 2) show how to use this to implement robust policy optimization under a prior and robust imitation learning; 3) demonstrate favorable comparisons with existing risk-sensitive and risk neutral algorithms for both settings. Right now I think that the description of the contributions hides the most useful contribution. Second, the paper was fairly notation and information dense. I recognize this is hard to account for, but I think it is worth revising the paper from the standpoint of remove as much notation as possible and including frequent redefintion of notation. I found myself flipping between initial definitions and subsequent use of notation a fair amount. As an example of potentially redundant notation, the paper uses $u$ for state frequencies and $\mu$ for expected feature counts and switches back and forth between use $R$, the state-by-state reward estimates and $W$ which are the weight estimates for reward functions. These are equivalent encodings, and so the paper might be simpler if the authors picked one. For example, consider the start of section 4 (l.172-178). The second sentence defines $\psi$ in a subclause (which is missing a "," at the start) and then defines the central optimization problem. These are both important concepts that should be defined on their own and highlighted to the reader. I would break this into many more sentences: "Let ... policies. Let ... functions. Define a performance measure $\psi$ as.... This is useful because.... With this definition, we can define our risk averse optimization: ...." In l.177, the authors use $\rho$ and rely on the reader remembering the definition from the start of the previous section. While this is somewhat small, I think it adds up over the course of the paper. Finally, I think the authors should move some of their examples further up in the paper. The paper describes risk aversion well, but I think that the authors could bring examples higher up in the paper to give an intuitive sense of why this is valuable. For example, including some parts of Fig. 4 earlier on to help motivate the reader to work through the notation will make the paper much more readable.

Relation to Prior Work: Yes, I think this is fairly clear. One area to improve is l.177, where is isn't clear that the state-occupancy distribution formulation of the LP comes from prior work. (In particular, instead of "We make use of the one-to-one..." consider "We build on the classic LP formulation of MDP planning, which optimizes the state occupancy distribution subject to flow constraints" or something similar.)

Reproducibility: Yes

Additional Feedback: Overall, I think this paper is an OK presentation and evaluation of a _really_ neat and potentially influential idea. I think that the paper should be accepted and I am open to revising my score upwards if the authors can make substantial improvements on presentation/evaluation. Update after discussion phase: my position is unchanged --- this is a good paper that should be accepted.


Review 2

Summary and Contributions: This paper describes a method for imitation learning in the paradigm of combining an IRL method with a planning method. It deviates from previous methods by using a risk-sensitive planning method that is adjustable in its risk sensitivity and hence more versatile than fully adversarial or fully risk-neutral approaches.

Strengths: Adjustable risk-sensitivity seems vital for many applications of imitation learning, and it is currently understudied. I found the example on the bottom of page 1 fairly convincing support for the claim that application-specific risk-sensitivity is valuable, in that is satisfies some real need on the part of the developer. The solution method is fairly clean and intuitive as a solution to a linear program. The experiments, while toy, do demonstrate the value of the approach over either extreme and over previous approaches. The paper is useful because of its careful analysis, strong motivation, and informative discussion of related work.

Weaknesses: The experiments are still a bit toy and small-scale. The paper would be significantly more convincing with human experiments or experiments in larger domains. I was not convinced that conditional value-at-risk is the best approach for adjustable risk-sensitivity, though it is fairly natural and has support in the previous literature. The key issue is how to handle misspecification of the Bayesian prior.

Correctness: As far as I can tell the methods follow well-known patterns and seem correct. The experimental design was convincing -- avoiding a focus on a particular base reward function keeps the analysis fair and clear. Minor nitpicks: l.137 "This is a strict generalization of the standard MDP formalism" - this seems an unnecessary and possibly confusing remark. The standard MDP formalism says nothing (as far as I know) about the form of the reward function R(s,a,s'). It need not be a table.

Clarity: Overall I found the paper very clear and well written, specifically: - The example at the bottom of the first page was very helpful in motivating the topic. - Table 1 was helpful in understanding the relation to other work. - The experimental analysis was clear. The choice not to analyze the effects with regard to a particular "true" reward function was effective in avoiding possible confusion. - Sorting the reward functions in Figure 5 was a good idea. Minor comments: l.25 demonstrator’s values -> demonstrator’s preferences [or, objectives] l.28 "to learn" -> "it to learn" l.83 "Bayesian IRL infers a posterior distribution over which rewards are most likely, but only optimizes for the mean" - this might be true in these papers, but obviously it is not the true Bayesian way or the true meaning of "Bayesian IRL" l.107 GAIL -> and GAIL l.110 between -> among l.336 it’s -> its References: many problems with over- and under-capitalization of titles, names, acronyms (Aaai, Icml), etc. NIPS-> NeurIPS Source for [25]?

Relation to Prior Work: I think this was very well done overall. The care taken with comparative analysis is impressive. The distinction between this paper and [8] "Efficient probabilistic performance bounds for inverse reinforcement learning." could be emphasized a bit more. When it was first mentioned the difference seemed minor because I did not realize the significant difference between value at risk and conditional value at risk, nor the stark difference in the ability to optimize for these metrics. Making this more explicit on line 95 could avoid readers believing that this is a small change on previous work. There is a bit of recency bias: - The older citations for imitation Learning at the beginning should be added. Imitation learning didn't start in 2018. - Risk-sensitive planning under uncertainty is much older than the 2010s. See, e.g., S. Koenig and R. G. Simmons. Risk-sensitive planning with probabilistic decision graphs. In Proceedings of the Fourth International Conference on Principles of Knowledge Representation and Reasoning (KR-94), pages 2301–2308, 1994. - Perhaps look at older work on robust RL (as distinct from solving robust MDPs), noting that the standard RL update rules are certainty-equivalent. See, e.g., Jun Morimoto, Kenji Doya, Robust Reinforcement Learning NIPS 2001 and Neural Computation 2005.

Reproducibility: Yes

Additional Feedback:


Review 3

Summary and Contributions: This paper studies imitation learning through a Bayesian perspective. Previous work mostly focuses either in risk-neutral IRL (by expected reward maximization) or through a minmax perspective (adversarially). This paper proposes a new objective for learning: weighted combination of expected return and conditional value at risk. Under linear reward assumption, the objective reduces to a linear program and could be solved efficiently. However, this needs a prior or posterior of the reward function as an input. In summary, this work provides a bridge between the previous two extreme frameworks, and achieves better empirical performance on some simple domains.

Strengths: I like the idea of balancing the risk and return in the imitation learning objective, using a hyper-parameter $\lambda$. The algorithm is well-stated, and under linear reward assumption, the method reduces to a simple LP and is computationally efficient. The empirical performance is also examined in two simple domains, to illustrate how $\lambda$ guides the learned policy from risk-neutral (but potentially high variance) to risk-averse. Also, in these two domains, the proposed method achieves better learning performance than baseline methods.

Weaknesses: The method is easy to understand and implement, however I feel the success of the method relies heavily on the following assumptions, which the authors do not justify well: - The method seems rely heavily on the quality of prior/posterior distribution of the reward, any justification (theoretically/empirically) how the method performs under misspecification of the reward distribution? - The linear structure of the reward brings computational convenience, however it is hard for reward to satisfy this structure in real-world applications. Any justification for this? Or empirically examine how the method performs under some complex domains? - The optimization objective seems have SA variables in total, which becomes huge in large state space. Any generalization of the method when we could parametrize the policy? Also, another concern from the computationally perspective: it may not be easy to get a prior/posterior distribution in real-world applications, it may affect the use of the proposed method. Empirically, the authors examine the performance of the model in two simple domains, and it will be great to see more experiments from some complex domains in deep RL domain or health care domain.

Correctness: The proposed algorithm is correct and the empirical methodology seems reasonable.

Clarity: The paper is well-written and easy to follow. One minor improvement may be incorporate more information about how to get the reward distribution and the complexity of it, to make the paper more self-contained.

Relation to Prior Work: The paper gives clear comparison with previous imitation learning method in Table 1. However I am not that familiar with the bayesian imitation learning literature, and could not give a judgement about how thorough it is.

Reproducibility: Yes

Additional Feedback: The comments and suggestions are shown in the "Weakness" part, and I am willing to change my score if they are well-justified.


Review 4

Summary and Contributions: This paper presents Bayesian Optimization for Imitation Learning (BROIL). The main motivation for the method is to learn a policy that is robust with respect to uncertainty that arises from inverse reinforcement learning. BROIL is framework for optimization of a policy that minimizes baseline regret given demonstrations from experts. They compare their approach to existing risk sensitive inverse reinforcement learning algorithms and claim improvements in expected return.

Strengths: Overall the paper is sound, straightforward, and meets its well-stated objectives. - The paper is easy to read, and the work has been well situated in the context of other related work. - The BROIL formulation is natural to the extent that I am surprised it is indeed novel in the context of RL. - There’s a good potential to extend this work to more complex domains

Weaknesses: It’s hard to assess the implications of this work. There are many possible claims of novelty: - This is the first work to extend adapt the concept of conditional value at risk to the inverse reinforcement setting - The BROIL formulation, which seems to be just a linear combination of expected reward with CVar - The fact that CVar and BROIL can be formulated as a linear program. Which of these are the claims of the paper? The second worry is that the scope of the method is unclear, and consequently, so is the utility of the method in practice. The example are small, which is not a problem in its own right, but we do not get an intuition about the limiting factors. Is it the in the loop MDP? The method is parameterized on having samples from the posterior, which they generate using MCMC. Assuming the samples are not exact, how do approximations in the MCMC method propagate onto optimization of BROIL? The method is presented in the context of inverse RL. Has it already been addressed within ordinary (non-inverse) RL? There are several examples of RL problems where taking risk into account is critical. In some sense, the inverse RL/imitation learning seems a bit superfluous to their method.

Correctness: They appear correct.

Clarity: The paper is clearly written and well structured.

Relation to Prior Work: The paper makes clear it’s relation to existing work.

Reproducibility: Yes

Additional Feedback:

[Author Response · NeurIPS 2020]

We would like to thank the reviewers for their detailed comments and suggestions.

**Scalability:** As noted by Reviewer 1, our method scales well to much larger domains. For example, our method can
optimize a robust policy for a machine replacement problem with 5,000 states in only 163 seconds. We can optimize
a robust policy for a 60-by-60 gridworld (3,600 states) in under two minutes. For comparison, a state-of-the-art
CVaR optimization approach for MDPs with no uncertainty over the reward function takes 2 hours for a similar-sized
gridworld (see [1] Section 5, last paragraph). We will add experiments demonstrating the scalability of our method
to the appendix and will add an experiment where we transfer a learned reward function to a new environment (as
suggested by Reviewer 1).

**Reviewer #1**
> *Right now I think that the description of the contributions hides the most useful contribution.*
Thank you for the suggestion, we will clarify the contributions as suggested.

> *The robust implementation of the method (4.d) doesn't match the demonstration for one state.*
The reason is that Bayesian IRL does not assume demonstrator optimality, only Boltzman rationality. We used a
relatively small inverse temperature resulting in reward function hypotheses that allow for occasional demonstrator
errors. Using a larger inverse temperature will cause the robust policy to match all the demonstrator's actions.

> *The paper is fairly well written, but there is room for improvement in the paper presentation.*
Thank you for the detailed and constructive suggestions. We will make the notation more consistent (for example by
sticking to $\mu$ and $w$ as much as possible) and add notation reminders throughout the paper. We will also introduce the
examples as a motivation earlier in the paper.

**Reviewer #2**
> *Is conditional value-at-risk the best approach for adjustable risk-sensitivity?*
We used CVaR because of its popularity and interpretability, but it is true that it is not always the best metric. BROIL
actually works with any convex risk measure, such as EVaR or entropic risk, the only modification is that the linear
program would need to be replaced by a convex optimization problem. We will make this clear in the paper.

> *The key issue is how to handle misspecification of the Bayesian prior.*
This is a good point and something we would really like to tackle in a followup work. The Bayesian statistics community
has devoted a lot of effort to addressing this problem; we will add appropriate pointers.

**Reviewer #3**
> *The method seems to rely heavily on the quality of prior/posterior distribution of the reward . . .*
Yes, this is true for all Bayesian methods.

> *The linear structure of the reward brings computational convenience, however it is hard for the reward to satisfy this*
*structure in real-world applications.*
We agree that linear approximation methods (including linear regression, and linear value function approximation) have
limits, but their simplicity, speed, and generally smaller data needs (bias-variance tradeoff) make them often very useful.

> *Any generalization of the method when we could parametrize the policy? . . . deep RL or healthcare experiments?*
These are good suggestions. The BROIL objective is convex and nearly everywhere differentiable so it could also be
used in place of expected return in a policy gradient-style approach. We judged this to be beyond the scope of this paper,
but will mention this idea in the paper as an important and interesting area for future work.

**Reviewer #4**
> *Assuming the samples are not exact, how do approximations in MCMC propagate onto optimization of BROIL?*
Thank you, this is an important question. It has been investigated in the stochastic programming community in the
context of the SAA method. We will include pointers to relevant literature on this topic in the revision.

> *The method is presented in the context of inverse RL. Has it already been addressed within ordinary (non-inverse) RL?*
Several similar methods have been studied in the context of RL, but the key difference is that most RL work considers
uncertain transition probabilities, while in IRL it is rewards that are uncertain. This difference has a major impact on
the type of algorithms that are appropriate for the two settings. Most relevant papers in ordinary RL, which address
robustness/risk aversion to *model error*, are distributionaly robust MDPs (Xu & Mannor 2012), percentile optimization
(Delage & Mannor 2010), and epistemic risk aversion (Eriksson & Dimitrakakis 2019).

## References

[1] Yinlam Chow, Aviv Tamar, Shie Mannor, and Marco Pavone. Risk-sensitive and robust decision-making: a cvar optimization
approach. In *Advances in Neural Information Processing Systems*, pages 1522–1530, 2015.


[Meta-Review · NeurIPS 2020]

This is a nice paper on robust/safe imitation learning. Could be better to link with Bayesian safe exploration for RL. After further discussion, all reviewers agree that it should be accepted.